# Dietary Intake of Green Nut Oil or DHA Ameliorates DHA Distribution in the Brain of a Mouse Model of Dementia Accompanied by Memory Recovery

**DOI:** 10.3390/nu11102371

**Published:** 2019-10-04

**Authors:** Emiko Takeyama, Ariful Islam, Nakamichi Watanabe, Hiroe Tsubaki, Masako Fukushima, Md. Al Mamun, Shumpei Sato, Tomohito Sato, Fumihiro Eto, Ikuko Yao, Takashi K. Ito, Makoto Horikawa, Mitsutoshi Setou

**Affiliations:** 1Department of Food Science and Nutrition, Graduate School of Human Life Sciences, Showa Women’s University, 1-7-57 Taishido, Setagaya-ku, Tokyo 154-8533, Japan; takeyama@swu.ac.jp (E.T.); nakamich@swu.ac.jp (N.W.); 2Institute of Women’s Health Sciences, Showa Women’s University, 1-7-57 Taishido, Setagaya-ku, Tokyo 154-8533, Japan; 3Department of Cellular and Molecular Anatomy, Hamamatsu University School of Medicine, 1-20-1 Handayama, Higashi-ku, Hamamatsu, Shizuoka 431-3192, Japan; ariful222222@gmail.com (A.I.); amamun5245@gmail.com (M.A.M.); shumpe.sato@gmail.com (S.S.); titon0620@gmail.com (T.S.); etofmhr@gmail.com (F.E.); itotakakun77777@gmail.com (T.K.I.); mh.alkerus@gmail.com (M.H.); 4The Institute of Statistical Mathematics, 10-3 Midori-cho, Tachikawa-si, Tokyo 190-8562, Japan; tsubaki@ism.ac.jp; 5International Mass Imaging Center, Hamamatsu University School of Medicine, 1-20-1 Handayama, Higashi-ku, Hamamatsu, Shizuoka 431-3192, Japan; yaoik5@gmail.com; 6Department of Optical Imaging, Institute for Medical Photonics Research, Preeminent Medical Photonics Education & Research Center, Hamamatsu University School of Medicine, 1-20-1 Handayama, Higashi-ku, Hamamatsu, Shizuoka 431-3192, Japan; 7Department of Systems Molecular Anatomy, Institute for Medical Photonics Research, Preeminent Medical Photonics Education & Research Center, 1-20-1 Handayama, Higashi-ku, Hamamatsu, Shizuoka 431-3192, Japan

**Keywords:** DHA, green nut oil, Alzheimer’s disease, dementia, SAMP8 mouse, DESI-IMS, memory efficiency

## Abstract

Docosahexaenoic acid (DHA), an omega-3 polyunsaturated fatty acid, has significant health benefits. Previous studies reported decreased levels of DHA and DHA-containing phosphatidylcholines in the brain of animals suffering from Alzheimer’s disease, the most common type of dementia; furthermore, DHA supplementation has been found to improve brain DHA levels and memory efficiency in dementia. Oil extracted from the seeds of *Plukenetia volubilis* (green nut oil; GNO) is also expected to have DHA like effects as it contains approximately 50% α-linolenic acid, a precursor of DHA. Despite this, changes in the spatial distribution of DHA in the brain of animals with dementia following GNO or DHA supplementation remain unexplored. In this study, desorption electrospray ionization imaging mass spectrometry (DESI-IMS) was applied to observe the effects of GNO or DHA supplementation upon the distribution of DHA in the brain of male senescence-accelerated mouse-prone 8 (SAMP8) mice, a mouse model of dementia. DESI-IMS revealed that brain DHA distribution increased 1.85-fold and 3.67-fold in GNO-fed and DHA-fed SAMP8 mice, respectively, compared to corn oil-fed SAMP8 mice. Memory efficiency in SAMP8 mice was also improved by GNO or DHA supplementation. In summary, this study suggests the possibility of GNO or DHA supplementation for the prevention of dementia.

## 1. Introduction

Docosahexaenoic acid (DHA), one of the most important omega-3 (ω -3) polyunsaturated fatty acid (PUFA) found in the brain, is essential for the development and function of the brain [1,2]. Dietary supplementation of DHA has attracted significant interest in the last few decades for its several health benefits [3]. DHA has been linked to a lower risk of several prevalent diseases, including cancer, cardiovascular disorders, and Alzheimer’s disease (AD) [4,5,6]—one of the most common types of dementia and the most common neurodegenerative disorder in elderly people. In 2018, the number of patients affected by AD in the U.S. alone was estimated to be 5.7 million [7]. The major pathological hallmarks of AD are the deposition of amyloid β (Aβ) plaques and hyperphosphorylated tau in the brain [8]. In addition, Aβ is the main determinant of neuronal loss and synaptic disruption in the development of dementia [9].

Previous studies have reported decreased levels of linoleic acid (LA; C18:2), oleic acid (OA; C18:1), arachidonic acid (AA; C20:4), eicosapentaenoic acid (EPA; C20:5), docosapentaenoic acid (DPA; C22:5), DHA (C22:6), and DHA-containing phosphatidylcholines in the brain of patients with dementia [2,10,11]. DHA supplementation increases levels of DHA in the brain and alleviates the Aβ load, tau phosphorylation, neuroinflammation, and neuronal apoptosis [12,13,14,15]. There is, however, a paucity of data on changes in the spatial distribution of free fatty acids (FFAs) in the brain after dietary intake of DHA. Thus, the present study was conducted in order to evaluate changes in the spatial distribution of DHA and other FFAs associated with dementia in the brain of senescence-accelerated mouse-prone 8 (SAMP8) mice—a mouse model of dementia, after dietary supplementation with DHA [16,17]. SAMP8 mice display many features of dementia, such as oxidative damage, degeneration of the brain stem, neuronal cell death, Aβ alternation, tau hyperphosphorylation, blood-brain barrier dysfunction, progressive learning, and memory deficits [16,17].

Additionally, the effects of green nut oil (GNO), also known as sacha inchi oil—oil extracted from the seeds of the evergreen climbing plant *Plukenetia volubilis*, were investigated in SAMP8 mice. GNO contains approximately 50% α-linolenic acid [18,19]. Human and other mammals can synthesize DHA from α-linolenic acid through a progressive series of enzymatic elongation and desaturation reactions [2,3]. As a result of the well-known health benefits of DHA, the continuing demand for DHA containing fish oil has exerted a substantial impact upon marine populations; thus, it has been suggested that mass scale fishing is no longer sustainable [20]. GNO is expected to be a sustainable alternative source of DHA instead of fish oil DHA. Moreover, GNO is less expensive compared to DHA and does not require special instruments for extraction and purification.

Over the last few decades, imaging mass spectrometry (IMS) has become a powerful tool for the direct visualization, identification, and analysis of biomolecules in proteomics and metabolomics [21,22]. By using IMS, several FFAs, phospholipids, peptides, and other metabolites have been identified in a number of tissues [23,24,25,26]. A recently developed IMS technique, known as desorption electrospray ionization imaging mass spectrometry (DESI-IMS), has emerged as a rapid, nondestructive, and powerful IMS technique for the detection of FFAs, phospholipids, proteins, drugs, metabolites, and other small molecules at µm scale resolution under ambient condition with the advantage of minimal sample preparation [27,28,29].

In this study, DESI-IMS was applied in order to analyze changes in the spatial distributions of FFAs, including DHA, in the brain of SAMP8 mice after supplementation with GNO or DHA. The Y-maze test was performed to investigate the effects of GNO or DHA supplementation upon the behavioral phenotype of SAMP8 mice.

## 2. Materials and Methods 

### 2.1. Animal

Thirteen weeks old male SAMP8 mice were purchased from Sankyo Labo Service Corporation, Inc. (Tokyo, Japan) and reared in individual cages at 22 ± 2 °C with a light/dark cycle of 12 h (light period: 7 am to 7 pm). Food and tap water were provided ad libitum. During the first week of the preparatory rearing periods, the diet consisted of CRF-1 (Charles River Formula-1) pellets from Charles River International Laboratories, Inc. (Kanagawa, Japan). During the subsequent rearing period, the diet consisted primarily of AIN-93G (American Institute of Nutrition 93-G) from Oriental Yeast Co., Ltd., Tokyo, Japan. After the 1st week of preparatory rearing periods, the mice were divided into three groups. Group I (CO-fed) was comprised of SAMP8 mice that were fed a diet consisting of AIN-93G, in which soybean oil was replaced with 7% corn oil (CO). Group II (GNO-fed) was comprised of SAMP8 mice, fed on a diet consisting of AIN-93G, in which soybean oil was replaced with 7% GNO. Group III (DHA-fed) comprised of SAMP8 mice that were fed a diet consisting of AIN-93G, in which soybean oil was replaced with a mixture of 7% fish oil DHA-46 and CO (DHA-46:CO = 4:3).

CO, GNO, and DHA-fed mice were reared for 14 weeks. After 14 h of fasting and 2 h of water deprivation on the last day of rearing, mice were euthanized by decapitation and dissected. Following removal, the brain samples were stored at −80 °C until further analysis. This study was approved by the Showa Women’s University Animal Research Committee and conducted according to the regulations of Showa Women’s University Animal Research Committee (ethical permission number: 17-04), and Animal Care and Use Committee of the Hamamatsu School of Medicine.

### 2.2. Chemicals and Reagents 

Fish oil DHA-46 and GNO were provided by Tama Biochemical Co., Ltd. (Tokyo, Japan) and NPO Arcoiris Naturaleza (Matsudo, Chiba, Japan), respectively. Corn oil was purchased from Oriental Yeast Co., Ltd. (Tokyo, Japan). LC/MS grade methanol, 2-propanol, and ultrapure water were purchased from Wako Pure Chemical Industries (Osaka, Japan). OA, lysophosphatidylcholine (18:1), phosphatidylcholine (18:1/18:1), DHA, phosphatidylcholine (22:6/22:6), and sodium formate were purchased from Sigma-Aldrich (St. Louis, MO, USA). AA was purchased from Cayman Chemical (Ann Arbor, Michigan, MI, USA).

### 2.3. Group Size and Samples

For behavioral analysis, group sizes were as follows: DHA, *n* = 10; GNO, *n* = 9; and CO, *n* = 9. For DESI-IMS analysis, group sizes were as follows: DHA, *n* = 3; GNO, *n* = 3; and CO, *n* = 3. For DESI-IMS analysis, three mice were selected from each group by stratified sampling. All nine mice taken for DESI-IMS analysis were divided into three experimental sets by blindly taking one mouse from each group for each experiment.

### 2.4. Tissue Preparation

Following collection, the brains were rapidly frozen in powdered dry ice and stored at −80 °C. The tissues were mounted on a sample holder using optimal cutting temperature (OCT) compound (Sakura Finetek Japan, Tokyo, Japan) and sectioned at 10 μm thickness along the sagittal axis using a cryostat system (CM1950; Leica Biosystems, Wetzlar, Germany) at −20 °C on un-coated glass slides (Matsunami, Osaka, Japan, size: 76/26 mm; thickness: 0.8 to 1 mm) for DESI-IMS analysis. Hematoxylin and eosin (H&E) staining was used in order to facilitate histopathological examination.

### 2.5. DESI-IMS 

Brain sections were kept at room temperature to dry before DESI-IMS acquisition. Mass spectra were acquired in negative ion modes. All experiments were performed using a DESI source attached to a quadrupole time-of-flight (Q-TOF) mass spectrometer (Xevo G2-XS Q-TOF, Waters, Milford, MA, USA). The mass spectra were calibrated externally using sodium formate solution (500 µM) in a 90:10 ratio of 2-propanol: water (v/v), prior to measurement. The spray solvent (98:2; methanol: water, v/v) was delivered at a flow rate of 2 µL/min using a solvent pump (ACQUITY UPLC Binary Solvent Manager, Waters, Milford, MA, USA). Ions from brain tissues were obtained in a mass range of *m/z* (mass-to-charge ratio) 100 to 1200 Da. All other parameters used in DESI-IMS are listed in Table 1.

With the exception of *m/z* 303.23 and 327.23, to assign all other candidate molecules corresponding to each targeted *m/z*, the Human Metabolome Database (http://www.hmdb.ca/spectra/ms/search) for mass accuracy and biological distribution was used alongside previous reports [30]. Tandem mass spectrometry (MS/MS) was performed in order to confirm molecular ions against *m/z* 303.23 and 327.23 on the same instrument using collision energy 10 eV, source temperature 120 °C, capillary voltage 4 kV, and 98% methanol as solvent at a flow rate 2 µL/min. To confirm whether all detected molecular ions corresponded to fatty acids (FAs) were from FFAs or FA fragments from lipids, standard OA, OA containing lysophosphatidylcholine (LPC), OA containing phosphatidylcholine (PC), DHA, and DHA containing PC were analyzed using same instruments and parameters as mentioned in Table 1.

### 2.6. DESI-IMS Data Analysis

MassLynx (Waters, Milford, MA, USA; version 4.1) software was used for data acquisition and processing, HDImaging (Waters, Milford, MA, USA; version 1.4) and IMAGE REVEAL (Shimadzu, Kyoto, Japan; version 1.0.1.8345) software was used for image analysis. IMAGE REVEAL was used particularly for the analysis of the intensity of each *m/z* at each pixel.

### 2.7. Mouse Behavior

Behavioral tests were conducted using the Y-maze test, with slight modifications to the method reported by Yamada et al. [31]. Briefly, mice were placed inside a Y-shaped device. Three arms (A, B, and C) contained objects of different colors and shapes placed on the outer walls. Exploratory behavior of the mice was recorded. Y-maze trials were performed three times in total during the breeding period (at the age of 14 weeks, 22 weeks, and 27 weeks) for all mice in each group. Measurements were performed for 8 min. The arms that a mouse entered while freely exploring inside the maze were recorded in order as the "total arm entry number (spontaneous behavior amount)". The number of times a mouse selected different arms three consecutive times was calculated as the "number of alternating behaviors". “Replacement behavior ratio" was determined as follows to assess spatial/short-term memory:Alternating behavior rate (%) = number of alternating actions/(total arm entry number − 2) × 100. 
Efficiency = log (alternating behavior rate/number of alternating actions).

### 2.8. Statistical Analysis

All statistical analyses were performed using SPSS (version 16) and MS Excel (version 2019) software. All values are expressed as mean ± SEM (standard error of the mean). Differences were considered significant with *p* values less than 0.05 (one-way ANOVA with Tukey′s multiple comparisons). IMAGE REVEAL (Shimadzu, Kyoto, Japan; version 1.0.1.8345) software was used to test the distribution pattern (Gaussian distribution or not) of DHA in the brain of SAMP8 mice.

## 3. Results

### 3.1. Detection of Target Peaks (m/z) from DESI-IMS Spectra

DESI-IMS experiments in negative mode over 100–1200 *m/z* range were performed, and mass spectra from SAMP8 mouse brains were acquired in order to analyze changes in the lipoquality following GNO or DHA supplementation. Figure 1 indicates the representative mass spectra obtained from SAMP8 mouse brains, and Appendix A depicts assigned peaks from DESI mass spectrum. 

In total, seven ions of interest were observed across the three groups with *m/z* 279.23, 281.25, 301.22, 303.23, 327.23, 329.25, and 331.27; these ions corresponded to LA (C18:2), OA (C18:1), EPA (C20:5), AA (C20:4), DHA (22:6), DPA (22:5), and adrenic acid (AdA; C22:4) respectively. With the exception of AdA, all other assigned FAs are associated with dementia. Corresponding molecular ions for *m/z* 303.23 and 327.23 were confirmed by tandem mass spectrometry (Appendix A). All other candidates were assigned based on their *m/z* accuracy (Table 2) and previous reports [30].

Additionally, five lipid standards were analyzed using the same instruments and parameters which were used to acquire mass spectra from SAMP8 mouse brains. FFAs were detected efficiently by DESI-IMS, but no fragments for FAs from LPC (18:1), PC (18:1/18:1), and PC (22:6/22:6) were detected (Appendix A).

### 3.2. Effects of Dietary Intake of GNO or DHA on Brain Distribution of FFAs 

After 14 weeks of GNO or DHA supplementation to SAMP8 mice, the distribution of LA, OA, EPA, AA, DHA, DPA, and AdA (Figure 2) was analyzed. 

Increased levels of LA, OA, EPA, AA, DHA, and DPA in DHA-fed mice, and increased levels of LA, OA, EPA, and DHA in GNO-fed mice were observed compared to that in CO-fed mice (Figure 3 and Appendix A). Among the three experiments, an increase in the distribution of AA in GNO-fed mice compared to that of CO-fed mice was observed in two experiments (Appendix A). No significant difference was found in the distribution of AdA between all three groups (Figure 3). Furthermore, maximum fold increase (with respect to the intensity of current ions of each targeted *m/z*) in the distribution/level of DHA in both GNO-fed and DHA-fed groups was observed compared to those in the CO-fed group (Figure 3 and Appendix A). Compared to that in the CO-fed mice, the distribution of DHA in the brain of SAMP8 mice was increased by 1.85-fold and 3.67-fold in the GNO-fed and DHA-fed mice, respectively (Appendix A).

### 3.3. Effects of Dietary Intake of GNO or DHA on Brain Distribution of DHA

As the maximum fold changes were obtained in the distribution of DHA following dietary supplementation with GNO or DHA than that obtained with CO supplementation in SAMP8 mouse brains, the distribution of the signal intensity of DHA in the brains of SAMP8 mice (Figure 4A–C) was analyzed. It was found that the distribution of the signal intensity of DHA was not Gaussian (Figure 4B). However, significant changes in the spatial distribution of DHA in the brains of GNO-fed or DHA-fed SAMP8 mice compared to that in CO-fed SAMP8 mice (Figure 4A,C and Appendix A) were observed. DHA distribution was significantly increased in the hippocampus in GNO-fed and DHA-fed mice relative to that in CO-fed mice (Figure 4A,C). Similarly, the dietary intake of both GNO and DHA also significantly increased the distribution of DHA in the cerebellum, cerebral cortex, thalamus, hypothalamus, and olfactory bulb in the GNO-fed and DHA-fed mice (Figure 4C, Appendix A, and Appendix A). Conversely, distribution of DHA in the septum was increased significantly only in DHA-fed mice but not in GNO-fed mice (Figure 4A,C). 

Spatial distribution of AA in the hippocampus of SAMP8 mice treated with CO, GNO, and DHA was also analyzed. In all three experiments, a significant increase in the distribution of AA in the hippocampus of GNO-fed and DHA-fed SAMP8 mice was found relative to that in CO-fed SAMP8 mice (Appendix A).

### 3.4. Effects of GNO or DHA on Memory Efficiency of SAMP8 Mice

The results of this investigation indicate that the dietary supplementation of GNO or DHA improved the memory efficiency of SAMP8 mice by 1.99-fold and 2.34-fold, respectively, compared to that of the CO-fed SAMP8 mice (Figure 5). 

## 4. Discussion

The results of this investigation suggest that the dietary supplementation with GNO or DHA could significantly increase the distribution of DHA in the brain of SAMP8 mice. The distribution of several other FFAs associated with dementia was also ameliorated in the SAMP8 mouse brain after GNO or DHA supplementation. To the best of our knowledge, this is the first report on the effects of GNO or DHA supplementation on the spatial distribution of DHA and other FFAs in the brain of the animal with dementia. Additionally, the possibility of the potential beneficial effects of GNO or DHA supplementation on memory efficiency in dementia using SAMP8 mice was revealed in this investigation.

In this study, DESI-IMS was applied in order to examine the spatial distributions of DHA and other FFAs associated with dementia in the brains of male SAMP8 mice. DESI-IMS is a recently developed technique that can be used to analyze the spatial distribution and identification of biomolecules, metabolites, and drugs [28]. Moreover, lipids and FFAs are easily ionized in DESI-IMS, and their metabolism lies at the core of many diseases, including dementia and cancer [4,23,25,32]. Seven molecular ions of FFAs were detected from the mass spectra of DESI-IMS, acquired in negative ion mode from brain slices of SAMP8 mice. Using MS/MS analysis, it was confirmed that AA and DHA corresponded to ions at *m/z* 303.23 and 327.23, respectively. Based on previous reports and mass accuracy, candidate molecules against the remaining five ions were assigned [30]. Analyzing lipid standards using the same instruments and parameters, which were used to acquire mass spectra from SAMP8 mouse brains, it was also confirmed that all fatty acids were detected as free fatty acids in this study.

A decrease in the total amount of LA, OA, EPA, AA, DHA, and DPA in the brains of humans and AD animal models—the most common form of dementia due to changes in the metabolism of FFAs—has been reported in previous studies [2,10,33]. Omega-3 PUFAs enhance fatty acid oxidation, increase fluidity and function of membrane proteins, regulate expression of genes related to signal transmission, serve as precursors to second messengers, and inhibit inflammation [5,34,35,36]. Using DESI-IMS, significant improvements in the distribution of LA, OA, EPA, AA, DHA, and DPA in the brains of SAMP8 mice after GNO or DHA supplementation were observed compared to CO supplementation. These results suggest that dietary supplementation with GNO or DHA in SAMP8 mice ameliorated the dementia associated deficiency of FFAs. GNO contains about 50% α-linolenic acid (precursor of DHA and EPA), 33% LA (precursor of AA), and 8% OA [3,18]. These fatty acids of GNO may have a role to improve the distribution of FFAs observed in the brain of GNO-fed SAMP8 mice in this study. DHA also can be converted to EPA and DPA through a process known as retroconversion and may help to improve DPA and EPA distribution in the brain of DHA-fed SAMP8 mice [37]. However, there is a paucity of data about the effects of DHA supplementation on the distribution or levels of LA and AA in animals. These FFAs play important physiological roles and can also be converted into other PUFAs and metabolites that are structural components of the cell membrane and crucial for several brain functions [3,10,36]. AA is one of the most abundant FA found in the brain. Although AA plays a notable role in inflammation, it acts as a precursor for signaling molecules and as a potent activator of gene transcription factors [1]. It also facilitates neurite overgrowth by stimulating syntaxin 3, which is critical for neuronal development [38].

Among all detected FFAs, the maximum increase in the distribution of DHA was found to occur in the brains of GNO-fed or DHA-fed SAMP8 mice. The distribution of DHA was increased by 1.85-fold and 3.67-fold in the brains of GNO-fed and DHA-fed SAMP8 mice, respectively, when compared to CO-fed SAMP8 mice.

To examine the distribution of DHA in the different regions of SAMP8 mouse brains, the regional-specific analysis was performed using images obtained from DESI-IMS for molecular ions of DHA at *m/z* 327.23. This study demonstrates that the dietary intake of GNO or DHA significantly augmented DHA distribution in most of the regions of SAMP8 mice brain. 

The accumulation of DHA in the hippocampus, the region where adult neurogenesis occurs and impacts memory function [39,40], in GNO-fed or DHA-fed SAMP8 brains was noted in this study. A decreased level of DHA was found in the hippocampus of SAMP8 mice in a previous study [41]. According to previous reports, DHA supplementation augments brain DHA levels, enhances neurogenesis, and reduces tau hyperphosphorylation, Aβ deposition, neurotransmission, and neuronal apoptosis [12,14,42]. Neurogenesis also occurs in the olfactory bulb and could be enhanced by increased levels of DHA in the olfactory bulb [43,44]. In this study, the increased distribution of DHA in the olfactory bulb of GNO-fed and DHA-fed SAMP8 mice was found compared to that in the CO-fed group. Additionally, increased levels of DHA in the cerebral cortex, cerebellum, hypothalamus, and thalamus were observed in SAMP8 mouse brains in both DHA-fed and GNO-fed groups compared to that of CO-fed SAMP8 mice. DHA is also expected to improve the functions of these brain regions. The cerebral cortex controls prospective memory and motor function [45], the cerebellum plays a significant role in goal-directed behavior and motor control [46,47], the thalamus plays important roles in memory [48], and the hypothalamus controls basic physiological functions, including emotion, reproduction, metabolism, energy expenditure, and flight responses [49]. After DHA, AA is the 2nd FFA in which the most significant change was found in DHA-fed and GNO-fed SAMP8 mice compared to CO-fed SAMP8 mice. An increased distribution of AA in the hippocampus of DHA-fed and GNO-fed SAMP8 mice compared to that of CO-fed SAMP8 mice was also observed in this study. LA found in GNO might increase the distribution of AA in the hippocampus of GNO-fed mice. However, to the best of our knowledge, there is a lack of previous reports about the effects of DHA supplementation on the distribution of AA in the animal brain. Further study is required to explore the mechanism by which DHA can affect the distribution of AA in the brain.

Dietary intake of DHA improves memory function in both experimental animal models and humans with dementia [50,51]. The recent meta-analysis study also demonstrated that DHA could improve memory function in healthy adults [52]. In order to explore the effects of dietary intake of GNO or DHA in memory efficiency, a Y-maze test, which is used to measure short-term memory and stereotypic behavior [53], was performed. It was observed that the memory efficiency of SAMP8 mice was improved by 1.99-fold and 2.34-fold in GNO-fed and DHA-fed mice, respectively, compared to CO-fed SAMP8 mice. All findings of this study suggest that both GNO and DHA supplementation improves brain DHA distribution and memory efficiency of SAMP8 mice. Currently, the increased demand for DHA containing fish oil, resulting from its tremendous health benefits, is proving to be detrimental to fish species and numbers [3]. This study suggests the prospective possibility of GNO as a sustainable source of DHA alternative to fish oil. However, further studies are required to investigate the role and mechanism of increased brain DHA following the dietary supplementation of GNO or DHA to improve memory efficiency and other pathological hallmarks of dementia in animal models and humans.

## 5. Conclusions

In summary, the dietary intake of GNO or DHA can ameliorate dementia-associated abnormal distribution of DHA in the brain and memory deficiency. Furthermore, GNO or DHA supplementation also can increase the distribution of other FFAs in the brain associated with dementia. Therefore, this study suggests the possibility of GNO or DHA supplementation for the prevention of dementia, and the potentiality of GNO as an alternative to fish oil DHA in the future.

## Figures and Tables

**Figure 1 nutrients-11-02371-f001:**
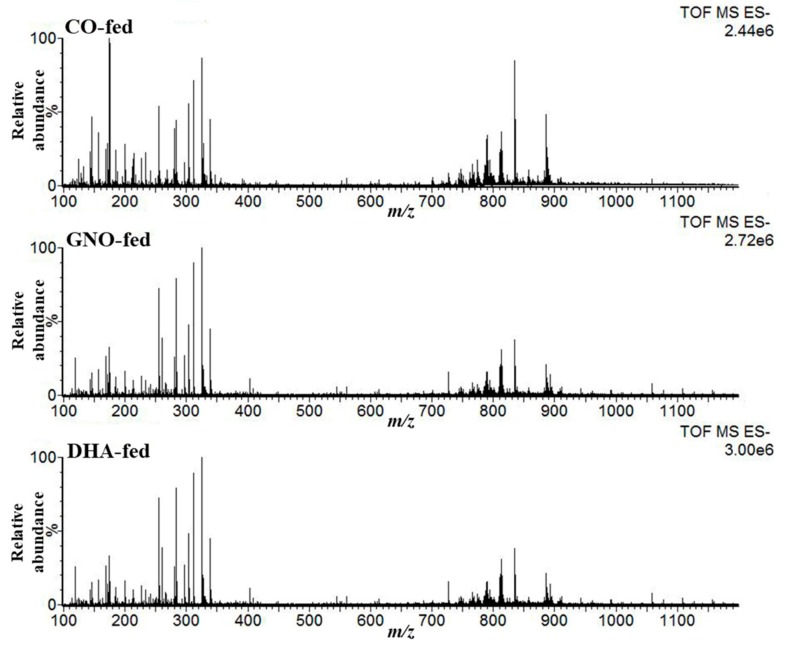
Representative mass spectrum of DESI-IMS at negative ion mode over *m/z* range 100–1200 Da acquired from sagittal slices of senescence-accelerated mouse-prone 8 (SAMP8) mouse brains. CO-fed, GNO-fed, and DHA-fed indicate corn oil-fed SAMP8 mice, green nut oil-fed SAMP8 mice, and docosahexaenoic acid-fed SAMP8 mice, respectively.

**Figure 2 nutrients-11-02371-f002:**
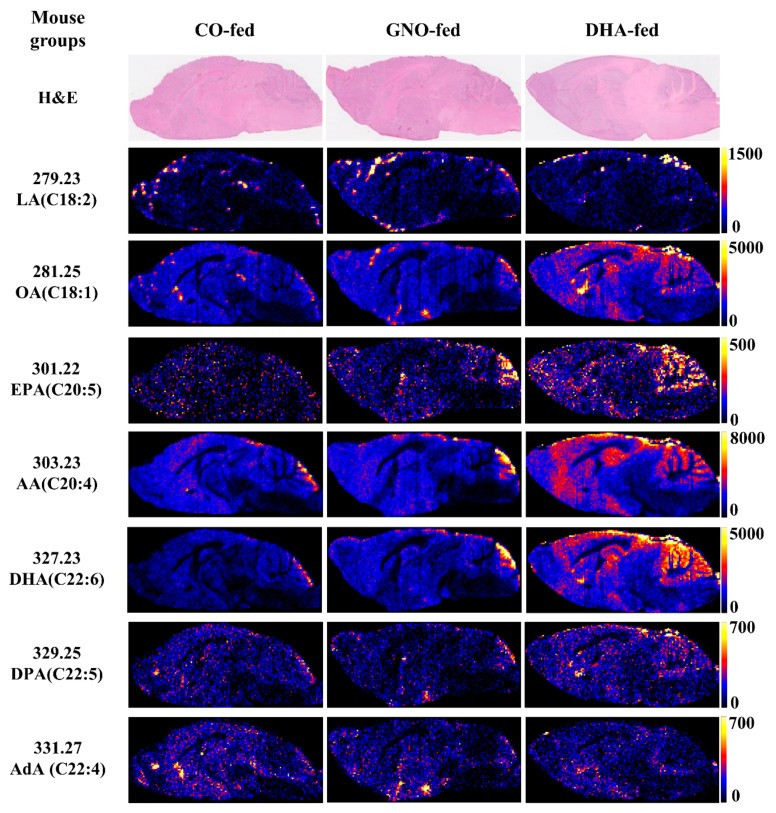
Selected molecular ion (M-H)**^-^** images of free fatty acids in sagittal slices of SAMP8 mouse brains from DESI-IMS. LA: linoleic acid, OA: oleic acid, EPA: eicosapentaenoic acid, AA: arachidonic acid, DHA: docosahexaenoic acid, DPA: docosapentaenoic acid, AdA: adrenic acid.

**Figure 3 nutrients-11-02371-f003:**
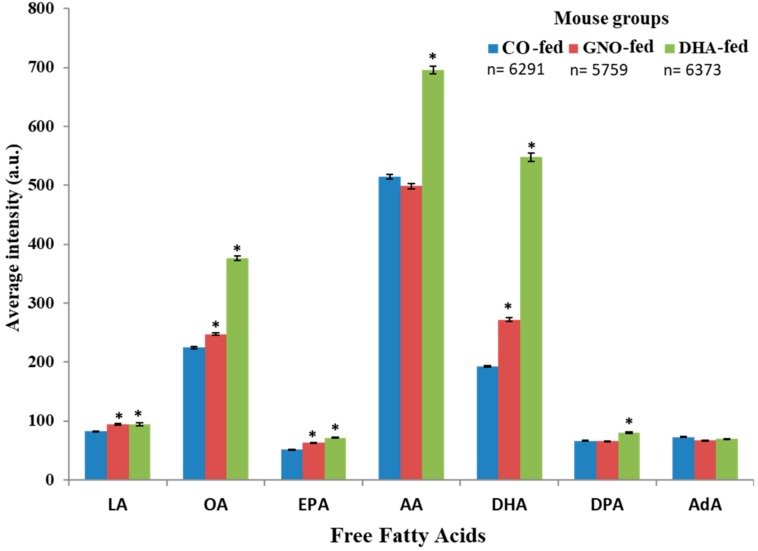
Average intensity (arbitrary unit; a.u.) of DHA and other free fatty acids in sagittal slices of SAMP8 mouse brains. All data are expressed as mean ± SEM. * indicates *p* < 0.001 compared to CO, and n indicates the number of pixels.

**Figure 4 nutrients-11-02371-f004:**
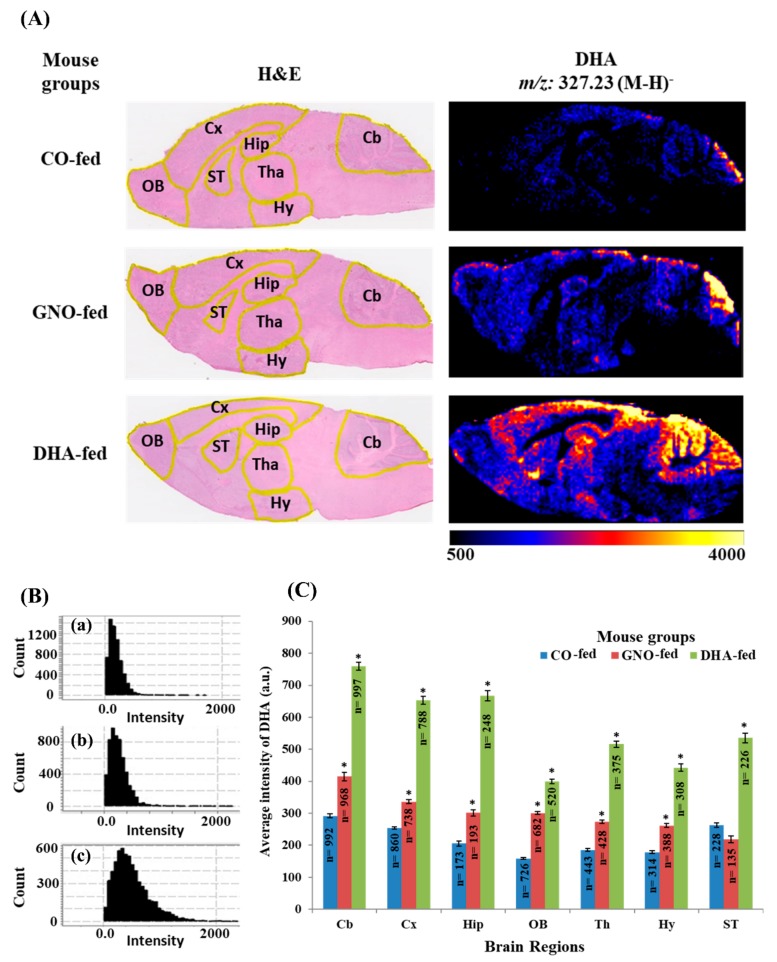
Distribution of DHA in different parts of the sagittal sections of SAMP8 mouse brains. CO-fed, GNO-fed, and DHA-fed indicate corn oil-fed SAMP8 mice, green nut oil-fed SAMP8 mice, and DHA-fed SAMP8 mice, respectively. (**A**) represents H&E stained brain slices and ion images of DHA. (**B**) represents the distribution of the signal intensity of DHA in SAMP8 mouse brains after supplementation with CO (**a**), GNO (**b**), and DHA (**c**). (**C**) represents average intensity (a.u.) of DHA in the different regions of SAMP8 mouse brains after supplementation with CO, GNO, and DHA. Here * indicates *p* < 0.001 compared to CO-fed SAMP8 mice. All values are expressed as mean ± SEM. Cb: cerebellum, Cx: cerebral cortex, Hip: hippocampus, OB: olfactory bulb, Tha: thalamus, Hy: hypothalamus, ST: septum, n: number of pixels.

**Figure 5 nutrients-11-02371-f005:**
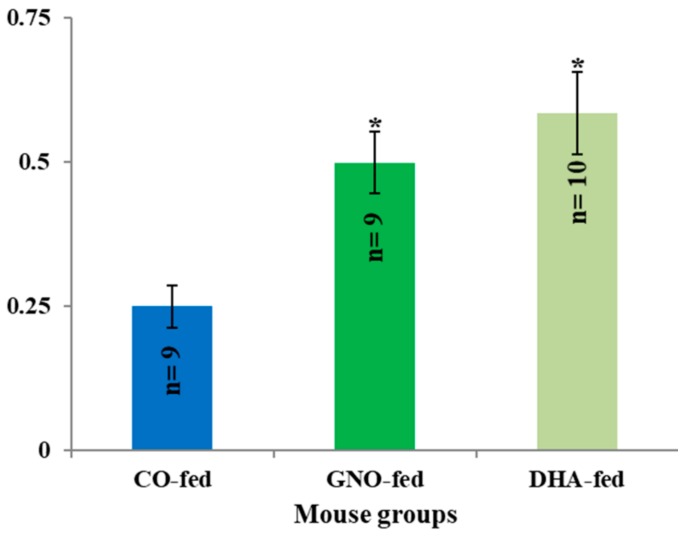
Effects of GNO and DHA on memory efficiency of SAMP8 mice. All data are presented as mean ± SEM. n indicates the number of mice, and * indicates *p* < 0.05 compared to CO-fed SAMP8 mice.

**Table 1 nutrients-11-02371-t001:** Parameters used for DESI-IMS analysis.

DESI Ion Source Parameters	Capillary Voltage	−4 kV
	Source Temperature	100 °C
	Spray Impact angle	80°
	Solvent	98% Methanol (v/v)
	Solvent Flow Rate	2 µL/min
	Nebulizing N_2_ Gas Pressure	0.4 MPa
DESI stage parameters	Pixel size	100 µM
	Scanning speed (X axis)	200 µM/sec
Data acquisition parameters	Polarity	−Ve
	*m/z* range	100 to 1200 Da
	Resolution	20000
	Mass window	0.02 Da
	Acquisition rate	1 spectrum sec-1
	Inlet voltage	4000 V

**Table 2 nutrients-11-02371-t002:** m/z accuracy of targeted molecules.

Free Fatty Acids	Theoretical *m/z*	Observed *m/z*	Mass Accuracy (ppm)
Linoleic acid (LA)	279.2330	279.2331	0.36
Oleic acid (OA)	281.2486	281.2487	0.36
Eicosapentaenoic acid (EPA)	301.2173	301.2170	1.00
Docosapentaenoic acid (DPA)	329.2486	329.2451	10.63
Adrenic acid (AdA)	331.2643	331.2639	1.21

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
