# Peer review of "Dietary Intake of Green Nut Oil or DHA Ameliorates DHA Distribution in the Brain of a Mouse Model of Dementia Accompanied by Memory Recovery"

_nutrients, 2019, doi:10.3390/nu11102371_

Round 1

Reviewer 1 Report

A very relevant manuscript with interesting imaging data assessing the impact of DHA and NGO supplementation in mice. The authors were rigorous with the laboratory methods, however I am concerned that the amount of evidence by itself does not support the conclusions. Perhaps toning down the implications and interpreting them within the context of other research in the topic would significantly improve the manuscript. 

Other specific comments: 

Abstract- the GNO clarification seems a bit out of place and takes from the flow of the abstract. I suggest moving it to the part where you say that DHA supplementation has been found to improve hippo DHA levels.

45- what do you mean by significant? Maybe important/physiologically active?

46- uniquely distributed does not mean anything

48- prodigious is a very strong and not very scientific word. I would tone it down and provide evidence of the claim. DHA is conditionally essential  (ALA has to come from the diet) and that is other reason why supplements are so popular.

55- clarify that these are polyunsaturated fatty acids

66- explaining the metabolism of ALA and DHA would make it easier to understand the role of GNO

Results

Why do you think there was also such a big increase in AA in the DHA group? This is important to discuss. Results seem to show that there was an increase in most fatty acids in the DHA mouse. Is this also something that has been reported before?

Discussion

258- There seems to be a great difference between the FFA in the GNO vs. DHA, especially in terms of AA, which does not normally respond to diet interventions but it seemed to respond to DHA in mice. Again, is there evidence that this is a common occurrence. It is difficult to tell with only one mouse per intervention, so that is a limitation that is important to acknowledge or substantiate with data from other studies.

297- do you know what was the distribution of AA in the hippocampus? Did it also increase in DHA or is this something that we would not expect based on the amount of AA in this region?

301- this should be in the results or methods. I am not sure how this statistics were performed with a sample of 1 per group?

The discussion is lacking and it is hard to appreciate the value of this study. I suggest making an effort to understand this results in light of previous evidence of the role of DHA and other fats distribution on the brain.

323- it is hard to make a judgement of the efficacy of the supplements improving memory because again, there was only 1 mouse per arm. Also, you first state that there was a strong correlation between DHA in the hippocampus and memory but then say that even though GNO did not modify this composition it had an effect on memory, which one is it? Similarly, DHA seems to have increased DHA everywhere by similar amounts, probably more on Cb, Cx, and Hip but the implication that it has to do with the hip is not supported beyond the fact that there is previous evidence that that region is associated with memory functions.

Conclusion

I disagree; if anything the results show that DHA was superior to GNO  

Reviewer 2 Report

Takeyama and colleagues present a work of interest regarding the effect of green nut oil and DHA dietary supplementation on brain distribution of several free fatty acid and on the behavior of SAMP8 mice. I acknowledge that the work done is very qualitative.

Nevertheless, I have some questions/remarks about this work:

Introduction

Could you explain better why you consider SAMP8 mice as a model of Alzheimer’s disease? These mice are generally described as a model of spongiform degeneration rather than a model of AD.

Material & methods

1. Why did you rear your mice in individual cages? They are sociable animals and it seems to me that they are in better conditions in little groups of several mice, specially for a pretty long study as the one done here.

2. What is the purpose of fasting and water deprivation before euthanasia?

3. What is the quantity of GNO or DHA consumed for each mice by day? Is this quantity transposable for human food supplementation?

4. At which age did you perform the behavior test? The section mention 3 times during the rearing period but more information should be provide. Moreover, do the repetition of the test change the result obtain, due to an habituation of the animals?

5. According to me the description of the behavior test and specially the calculation of the efficiency is unclear. What is a “correct answer” in this test? This must be clarified.

Results

1. For the figure 4A, authors say that “DHA distribution was significantly increased in the dentate gyrus of the hippocampus in GNO-fed and DHA-fed mice relative to that in CO-fed mice”. On Fig. 4A, the entire hippocampus is showed. The dentate gyrus should also be identified on brain slices but also quantified alone to permit this sentence

2. Results from the behavior test are difficult to understand, linked to the description of the test in the MM section.

Discussion

Authors suppose that LA might have a beneficial effect in AD through a decrease of inflammation. Is neuroinflammation a component found in SAMP8? If yes, it might be interesting to study this neuroinflammation.

Reviewer 3 Report

Docosahexaenoic acid (DHA), one of the most important omega-3 (ɷ-3) polyunsaturated fatty acids (PUFA) in the brain, has attracted significant interest in the last few decades for its health benefits and has been linked to a lower risk of several prevalent diseases, especially Alzheimer’s disease. In this manuscript, Dr. Setou and co-workers try to utilize desorption electrospray ionization imaging mass spectrometry (DESI-IMS) to observe the effects of GNO or DHA supplementation upon the distribution of DHA in the brain of a mouse model of dementia. The results demonstrate the dietary intake of GNO or DHA can ameliorate dementia associated abnormal distribution of DHA in the brain and memory, Furthermore, GNO or DHA supplementation also can increase the distribution of other FFAs in the brain associated with dementia, thus suggests the possibility of GNO or DHA supplementation for the prevention of dementia and potentiality of GNO as an alternative to fish oil DHA in the future. What’s more, the authors have made necessary corrections to improve paper quality.

The style and overall representation of the article are excellent. The authors have provided sufficient background and include all relevant references. The studies methods are properly described with accuracy and are suitable for Nutrients. I recommend this paper to be published in this journal.